# Proton enhanced dynamic battery chemistry for aprotic lithium–oxygen batteries

Yun Guang Zhu[1], Qi Liu[2], Yangchun Rong[2], Haomin Chen[1], Jing Yang[1], Chuankun Jia[1], Li-Juan Yu[3], Amir Karton[3], Yang Ren[2], Xiaoxiong Xu[4], Stefan Adams[1] & Qing Wang[1]

Water contamination is generally considered to be detrimental to the performance of aprotic lithium–air batteries, whereas this view is challenged by recent contrasting observations. This has provoked a range of discussions on the role of water and its impact on batteries. In this work, a distinct battery chemistry that prevails in water-contaminated aprotic lithium–oxygen batteries is revealed. Both lithium ions and protons are found to be involved in the oxygen reduction and evolution reactions, and lithium hydroperoxide and lithium hydroxide are identified as predominant discharge products. The crystallographic and spectroscopic characteristics of lithium hydroperoxide monohydrate are scrutinized both experimentally and theoretically. Intriguingly, the reaction of lithium hydroperoxide with triiodide exhibits a faster kinetics, which enables a considerably lower overpotential during the charging process. The battery chemistry unveiled in this mechanistic study could provide important insights into the understanding of nominally aprotic lithium–oxygen batteries and help to tackle the critical issues confronted.

[1] Department of Materials Science and Engineering, Faculty of Engineering, National University of Singapore, 117576 Singapore, Singapore. [2] X-Ray Science Division, Advanced Photon Source, Argonne National Laboratory, Argonne, Illinois 60439, USA. [3] School of Chemistry and Biochemistry, The University of Western Australia, 35 Stirling Highway Crawley, Perth, Western Australia 6009, Australia. [4] Ningbo Institute of Material Technology and Engineering, Chinese Academy of Sciences, Ningbo 315201, China. Correspondence and requests for materials should be addressed to Q.W. (email: msewq@nus.edu.sg).

The pursuit of high-energy power sources going beyond the state-of-the-art Li-ion batteries has evoked a surge of intensive studies of the lithium–air battery, as it has the potential of achieving nearly the same level of energy density as that of gasoline[1]. Although profound studies have been carried out, several technical challenges severely hinder the development of lithium–air batteries for practical application. Taking the most studied aprotic lithium–oxygen (Li-$O_2$) system as an example, the formation of insoluble and insulating lithium peroxide (Li$_2O_2$) during the discharge process leads to surface passivation and pore clogging of the cathode, which results in low round-trip energy efficiency and limited capacity[2–4]. Advances in electrocatalysts so far seem to have achieved only limited success in addressing the above issues. It remains a significant challenge that in a Li-air battery the oxygen reduction reaction (ORR) and oxygen evolution reaction (OER) take place electrocatalytically at the 'solid–solid' interface, which is intrinsically less favourable than those at the 'liquid–solid' interface in other metal-air batteries (or fuel cells)[5–9]. As such, soluble redox catalysts have recently been extensively investigated to transform the solid-state electrode reaction into a solution phase reaction[10–19]. Among the soluble OER catalyst, iodide received the most attention owing to its relatively good stability. Another critical issue for the aprotic Li-$O_2$ battery is that it is in essence an open system nominally, for which not only oxygen is fed into the battery upon operation; other species in air such as moisture are also inevitably introduced into the system. The presence of water in the electrolyte is generally considered to be detrimental as it attacks lithium metal at the anode and it may become involved in the ORR reaction at the cathode. For instance, water and protons were found in one study to significantly influence the crystal growth of Li$_2O_2$ (refs 20,21). In other studies, lithium hydroxide (LiOH) was however identified as the main discharge product in the presence of moisture[15,16], whereas disputes persist on the oxidation of LiOH by triiodide (I$_3^-$) during charging process[22–27]. Moreover, water was believed to catalyse the ORR reaction in aprotic Li-$O_2$ battery resulting in the formation of LiOH[28], and good cycling performance was achieved in humid $O_2$ (ref. 29). Therefore, owing to the complexity of the reaction, the battery chemistry of water-contaminated aprotic Li-$O_2$ cell remains to be elucidated[30].

Here we carefully investigate the influence of water on the battery chemistry of aprotic Li-$O_2$ cells when LiI is used as the OER redox catalyst. With the help of a Li$^+$-conducting ceramic membrane, we safely exclude any side-effects that may incur by the reactions of water and redox mediators with the lithium anode. One finding is that along with LiOH, lithium hydroperoxide (LiOOH) is detected to be one of the predominant discharge products, heralding a distinct battery chemistry for water-contaminated Li-$O_2$ batteries. As a rarely reported lithium compound, we study the crystallographic and spectroscopic characteristics of LiOOH both experimentally and theoretically, and find LiOOH presents much faster reaction kinetics towards I$_3^-$ as compared with Li$_2O_2$ and LiOH. A two-stage charging process is proposed in terms of the detailed studies to elucidate the mechanism of Li-$O_2$ cells involving water in the battery reactions.

## Results

**Identification and characterization of LiOOH.** Our study started off from the assessment of the reactivity of various discharge products of Li-$O_2$ batteries towards oxidation by I$_3^-$. Titrations of Li$_2O_2$ and LiOH with I$_3^-$ were firstly conducted in dimethoxyethane (DME) solutions. Despite the slightly more positive potential I$_3^-$ stays intact in Li$_2O_2$ suspension even after

stirring for 12 h (Supplementary Fig. 1), which can be explained by the sluggish reaction between the two species in accordance with our previous computational and experimental observation[17]. A similar phenomenon was observed here for LiOH, suggesting that in vigorously dried aprotic solution I$_3^-$ cannot be removed by Li$_2O_2$ and LiOH within the timescale of the titration. After adding $H_2O$ into the above Li$_2O_2$ suspension, the colour of I$_3^-$ was quickly bleached as a result of rapid reduction of I$_3^-$. In contrast, the presence of $H_2O$ in LiOH solution did not cause an appreciable change, whereas the bleaching happened instantaneously when $H_2O_2$ was added into the LiOH solution, although $H_2O_2$ itself was found to be stable with I$_3^-$ (Supplementary Fig. 1). To understand the above phenomena, we noticed the following two reactions for Li$_2O_2$ and LiOH have been reported, respectively[31,32]:

$$Li_2O_2 + H_2O \rightarrow LiOOH + LiOH \qquad (1)$$

$$LiOH + H_2O_2 \rightarrow LiOOH + H_2O \qquad (2)$$

In the presence of surplus water, the solid products of both reactions are expected to be in the hydrated form, LiOOH·$H_2O$ and LiOH·$H_2O$[33]. Interestingly, if not coincidentally, both reactions point to the same product—LiOOH, implying that the bleaching of I$_3^-$ might be induced by this compound. Though its crystal structure is not known, various authors suggest it to be the intermediate when producing Li$_2O_2$ by reactions of $H_2O_2$ with Li alcoholates in the corresponding alcohol (that is, methanol, ethanol and references therein)[34].

Hence, on the basis of the above titration tests, the reactivity of LiOOH is the highest and LiOH is the lowest towards oxidation by I$_3^-$. Whereas the comparison is arguably supported by the titrations, the formation of LiOOH, its reaction with I$_3^-$, and more importantly the existence of the compound in Li-$O_2$ battery, need to be unambiguously characterized and identified.

To discern the characteristics of LiOOH from LiOH and Li$_2O_2$, the three compounds were concertedly characterized by synchrotron X-ray diffraction (XRD), Raman and Fourier transform infrared (FTIR) spectroscopy measurements with attenuated total reflection (ATR) mode, for which both LiOH and Li$_2O_2$ were used as received, whereas wet powder of LiOOH was prepared by a simple reaction between LiOH and $H_2O_2$ in water/DME following reaction (2).

The structures of four different Li compounds were characterized by powder XRD (Supplementary Fig. 2). The diffraction pattern of LiOOH is clearly discriminated from those of LiOH, Li$_2O_2$ and Li$_2$O. In order to figure out the structure of the obtained LiOOH, high-resolution powder XRD was performed at 11-BM, Advanced Photon Source, Argonne National Lab (Fig. 1a). As seen from the LeBail fit results in Supplementary Fig. 3, the diffraction pattern of the LiOOH phase could be well indexed to a triclinic structure with $P1$ or $P\bar{1}$ symmetry and lattice parameters: $a = 6.3688$ Å, $b = 6.0878$ Å, $c = 3.2074$ Å, $\alpha = 79.598°$, $\beta = 101.832°$, $\gamma = 102.311°$, volume $= 117.69$ Å$^3$. The goodness of indexing, F(28), is as high as 795.1 with a zero-shift as small as $-0.0009°$, which together indicates that the fitting result is highly reliable. LeBail fits of an additional high-resolution XRD pattern for the $2\theta$ range up to 22° ($2\theta$ angle) showed result very close to those derived from the lower $2\theta$ angle range.

The crystal structure of LiOOH was solved from powder data starting from a comparison of the atomic arrangements of various compounds with related chemical compositions and reduced cells, noting that the close similarity of lattice parameters $a$ and $b$ as well as of $\beta$ and $\gamma$ suggests that the structure may be seen as a distorted variant of a monoclinic $C2/m$ or even an orthorhombic body-centred structure as originally proposed for LiOOH·$H_2O$ by Cohen[33]. Rietveld refinements of several of these starting

models converged to the structure shown in Fig. 2 with profile R and Chi$^2$ values ($R_{wp} = 7.86\%$, $R_p = 6.10\%$, $\chi^2 = 2.93$) closely approximating those of the model-free LeBail fit (Supplementary Fig. 3) clarifying that for the available data quality no alternative structure model can yield a significantly closer match. Accordingly, the bond valence sums (when using soft BV parameters)[35–37] of all atoms in the refined structure are close to the expectation value leading to a low global instability index of GII = 0.077 underlying the plausibility of the structure model. Geometry optimization of the result from the Rietveld refinement by DFT confirmed that this structure is metastable. Details of the structure parameters resulting from both the Rietveld refinement and the DFT geometry optimization are given in the supplementary material (Supplementary Tables 1 and 2). The structure consists of chains of H-bridged $OOH^-$ ions as well as of $Li^+$ ions arranged along the c-direction ($O-H\cdots O$ distances 1.12 and 1.46 Å from Rietveld refinement, or 1.07 and 1.46 Å from the DFT data), where each $Li^+$ is tetrahedrally coordinated by O atoms of two water molecules and two $HOO^-$ anions. The atomic arrangement is closely related to the one reported earlier for monoclinic $LiOH \cdot H_2O$ (Space group $C2/m$), as becomes more clearly visible when the primitive cell is used for that structure (see Fig. 2c). The structural similarity of $LiOOH \cdot H_2O$ and $LiOH \cdot H_2O$ also leads to similar stability. At 0 K the $LiOOH \cdot H_2O$ should according to the DFT calculations be marginally stable against the decomposition into $LiOH \cdot H_2O$ and ½ $O_2$, whereas at ambient conditions, the entropically favoured decomposition of $LiOOH \cdot H_2O$ proceeds easily. In the presence of $CO_2$ from ambient air $LiOH \cdot H_2O$ then reacts further to form $Li_2CO_3$ (Supplementary Figs 4 and 5).

The Raman spectrum shown in Fig. 1b for the $LiOOH \cdot H_2O$ sample reveals a distinct characteristic peak at around 860 cm$^{-1}$, assigned to the stretching of O-O bond based on the density functional theory (DFT) calculations (Supplementary Figs 6 and 7). In comparison, the O–O bond stretching of $Li_2O_2$ and $H_2O_2$ molecules is observed at ~790 and 877 cm$^{-1}$ (Supplementary Fig. 8), respectively, just straddling that of $LiOOH \cdot H_2O$. Other fingerprint peaks for $LiOOH \cdot H_2O$ were also observed at 80–150 cm$^{-1}$, implying $LiOOH \cdot H_2O$ is indeed a different species from LiOH and $Li_2O_2$. Characteristic IR responses of $LiOOH \cdot H_2O$ were also detected in the FTIR measurement (Supplementary Fig. 9), where the peak at 1,643 cm$^{-1}$ is identified as a $H_2O$ bending mode. Although other vibrations are not yet specified owing to a lack of reference data, the distinct spectra well evince $LiOOH \cdot H_2O$ as a new species relevant to Li-$O_2$ batteries.

**Electrochemical properties of LiOOH**. To investigate the catalytic effect of $I_3^-$ on the oxidation of the above lithium compounds, rotating disk electrode (RDE) was employed to probe the reactions, in which the powders of $LiOOH \cdot H_2O$, $Li_2O_2$ and LiOH were dispersed in LiI electrolyte. As shown in Fig. 3a, in the presence of $LiOOH \cdot H_2O$ suspension the limiting current for $I^-$ oxidation on RDE is enhanced by nearly 10 times as compared with those with LiOH and $Li_2O_2$. The direct oxidation of $LiOOH \cdot H_2O$ on RDE could be excluded as it generated almost zero current in the absence of LiI (Fig. 3a). Thus, such a considerable enhancement is rationalized by the catalytic reaction between the formed $I_3^-$ and LiOOH in the vicinity of RDE, which rapidly regenerates $I^-$. In contrast, the presence of LiOH or $Li_2O_2$ suspension has little influence on the reaction of $I_3^-$, consistent with the titration experiment.

The reactions of the various lithium compounds with $I_3^-$ were substantiated by battery charging test. The cell consists of a cathodic and an anodic compartment, which are separated by a piece of LAGP ceramic membrane (Supplementary Fig. 10a). The powders of $LiOOH \cdot H_2O$, $Li_2O_2$ and LiOH were loaded on the cathode (carbon felt) before it was fabricated into the cell. The use of $Li^+$-conducting membrane is crucial as it prevents $I_3^-$, water and oxygen from crossing-over and parasitically reacting with the Li metal in the anodic side. As shown in Fig. 3b, the theoretical charging time of $I^-$ to $I_2$ is ~7 h, whereas all the three cells present significantly longer charging process (>30 h), indicating the lithium compounds are involved in the reactions contributing to the charging capacity. In the absence of the above lithium compounds, the reactions of $I^-$ in DME electrolyte exhibit two distinct voltage plateaus at ~3.20 and 3.70 V, corresponding to the formation of $I_3^-$ and higher order polyiodide to eventually $I_2$, respectively (Supplementary Fig. 11)[38]. In comparison, the LiOOH cell reveals only one prolonged charging plateau at ~3.20 V, which is rational in terms of the titration experiment that the formed $I_3^-$ could instantaneously be reduced back to $I^-$ by LiOOH for extended charging, and the cell voltage is determined by the $I^-/I_3^-$ redox reaction on the electrode. So the overall reaction on the cathode only involves $I^-/I_3^-$-mediated oxidation of LiOOH, and the capacity is limited by the quantity of material loaded. In comparison, the $Li_2O_2$ cell presents two charging plateaus resembling that of the pure LiI cell (Supplementary Fig. 11), but with the second plateau greatly extended. Such a phenomenon has previously been observed in redox flow lithium–oxygen battery (RFLOB)[17] and is consistent with the titration that in aprotic electrolyte $I_3^-$ is unable to rapidly oxidize $Li_2O_2$, which requires a stronger oxidizer such as

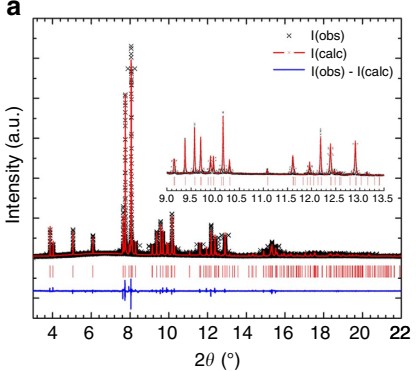

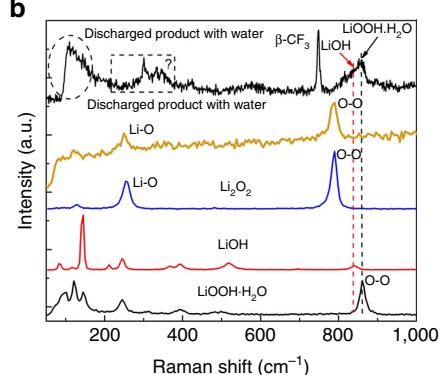

**Figure 1 | Characterizations of the various lithium compounds and the discharge product with or without water contamination. (a)** Rietveld refinement of the high-resolution X-ray diffraction pattern of the chemically synthesized $LiOOH \cdot H_2O$ phase. The wavelength is $\lambda = 0.41423 \pm 0.00004$ Å. The R factors of Rietveld refinement are $R_p = 6.10\%$, $R_{wp} = 7.86\%$, $\chi^2 = 2.93$. **(b)** Raman spectra of $Li_2O_2$, LiOH, $LiOOH \cdot H_2O$ and that of the discharge product collected from the cathode of a Li-$O_2$ cell without water or containing 9.1 vol.% $H_2O$ in the electrolyte.

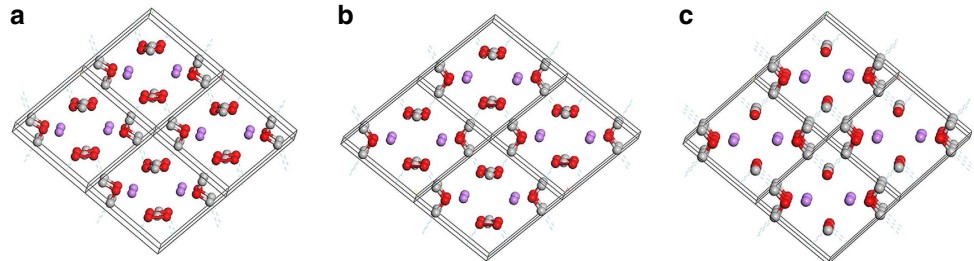

**Figure 2 | Comparison of crystal structures between LiOOH·H₂O and LiOH·H₂O.** Crystal structure of LiOOH·H₂O as derived from (**a**) Rietveld refinement of synchrotron X-ray powder data and (**b**) from DFT refinement. (**c**) The crystal structure of LiOH·H₂O given by Hermannson et al.[44] Here the primitive cell is shown to emphasize the close relation between the structures of both phases. (O: red; H: grey; Li: magenta). Broken lines indicate the hydrogen bonds stabilizing the structures.

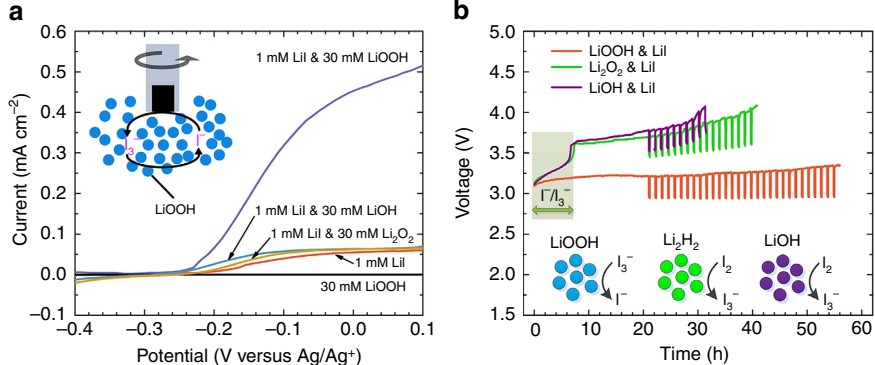

**Figure 3 | Electrochemical properties of LiOOH and the comparisons with LiOH and Li₂O₂.** (**a**) RDE measurements of 1 mM LiI in 0.1 M LiTFSI/DME with 30 mM LiOOH·H₂O, Li₂O₂ or LiOH dispersed in the solution. For comparison, the same measurements were conducted in the absence of LiI or lithium compounds suspension in the electrolyte. The rotating rate was 1,200 r.p.m. and the scan rate was 0.01 V s⁻¹. The inset illustrates the catalytic reaction between LiOOH and $I_3^-$ upon RDE measurement. (**b**) The charging curves of Li-LiOOH, Li-Li₂O₂ and Li-LiOH cells. Lithium foil was used as anode. LiOOH·H₂O precipitate, Li₂O₂ or LiOH powder in great excess to LiI in catholyte was loaded onto the cathode before the cells were assembled. The catholyte was 1 ml 0.5 M LiTFSI/DME containing 40 mM LiI. A LAGP membrane was used to segregate the two cell compartments. The cells were first charged at a constant current of 0.1 mA cm⁻² for 21 h, and then followed by GITT measurement (2 h charging at the same current plus 10 min resting).

I₂ formed at the higher voltage plateau. The charging of the LiOH cell is fairly similar to that of Li₂O₂, except for slightly larger overpotentials at the higher voltage plateau, presumably a result of sluggish reaction between LiOH and I₂, or more complex reactions[22–24].

To eliminate the overpotentials imposed by the membrane and other cell components during the charging process, galvanostatic intermittent titration technique (GITT) measurement was performed with the cells after 20 h charging. The relaxed cell voltage is on average ~2.95 and 3.55 V for the LiOOH and Li₂O₂ cells, and is slightly higher for the LiOH cell, broadly in agreement with the respective potential of $I^-/I_3^-$ and $I_3^-/I_2$ as determined by voltammetry (Supplementary Fig. 12). The above charging tests corroborate the previous comparison of the three compounds with $I_3^-$ and that LiOOH < Li₂O₂ < LiOH in terms of oxidation capability.

**Characterization of Li-O₂ battery with water contamination.** In order to verify the formation of LiOOH as an oxygen reduction product in moisture-contaminated Li-O₂ battery, water was deliberately introduced into the aprotic catholyte of Li-O₂ cells. As shown in Fig. 4a, the presence of water seems to be advantageous to the reduction of overpotential during the 10 h discharging process. With increasing water content in the electrolyte, the discharging plateau shifts upwards. The charging curves for all the cells predominantly present two voltage plateaus at ~3.50 V and 3.85 V, respectively. The lower voltage plateau is

assigned to the oxidation of $I^-$ to $I_3^-$, whereas owing to surface passivation of the electrode as generally observed in aprotic Li-oxygen batteries, the overpotential is considerably higher than that observed in Fig. 3b. The maximum charging time for $I^-$ to $I_3^-$ is ~4.5 h, so the extended charging would be a result of the catalytic reaction of $I_3^-$ with the discharge product, in which LiOOH was identified by Raman spectroscopy when probing the electrode after 10 h discharge (Fig. 1b).

However, the prolonged additional 4–5 h charging time at ~3.50 V could not account for the 10 h discharge, for which around half the discharge product seemingly remains intact with $I_3^-$. When the cells were further charged to a higher voltage, where the reaction of $I_3^-/I_2$ prevails, a second voltage plateau appeared at ~3.85 V with evidently extended charging for another 4–5 h. On the basis of the previous charging tests, either LiOH or Li₂O₂ may contribute to this process, whereas considering Li₂O₂ is instable in the presence of water, this extra capacity is deemed to be stemming from LiOH (or LiOH·H₂O). This is reasonable in terms of reaction (1), and that the discharge product of aprotic Li-O₂ battery, Li₂O₂, is converted into two distinct compounds co-existing in water-contaminated cells, of which LiOOH·H₂O reacts with $I_3^-$ at a lower voltage, whereas LiOH·H₂O reacts with I₂ at a higher voltage in a two-stage charging process. The presence of LiOOH and LiOH in the discharge product was confirmed by ATR-FTIR measurement (Supplementary Fig. 13), in which the characteristic peaks of LiOH and LiOOH·H₂O are clearly identified upon redox-assisted ORR reaction in the presence of water.

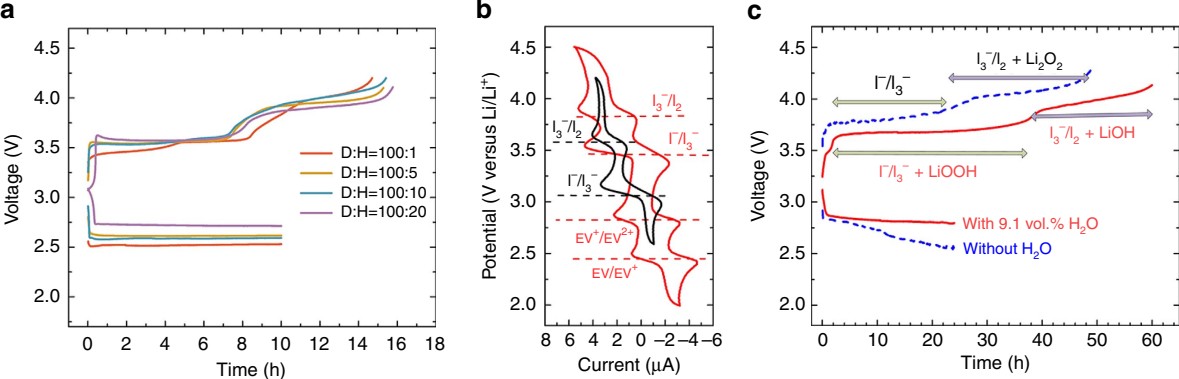

**Figure 4 | Electrochemical performance of Li-O₂ cells with different H₂O content in the catholyte.** (**a**) The charge–discharge curves of water-contaminated Li-O₂ batteries. The catholyte consisted of 0.5 ml 50 mM LiI and 0.5 M LiTFSI in DME. Different amount of water was introduced in the catholyte. D:H denotes the volume ratio of DME and H₂O. (**b**) The cyclic voltammograms of 2.5 mM LiI in 0.5 M LiTFSI/DME and 2.5 mM EVI₂ in 0.5 M LiTFSI/ DEGDME-DMSO (1:1 v/v). Pt disc and plate were used as the working and counter electrode, respectively. The scan rate was 0.02 V s⁻¹. (**c**) The charge–discharge curves of redox flow Li-O₂ battery using EVI₂ as the redox mediator. The catholyte consisted of 4 ml 15 mM EVI₂ and 0.5 M LiTFSI in DEGDME-DMSO (1:1 v/v) with or without 9.1 vol.% H₂O. The anolyte was 0.5 M LiTFSI in DEGDME. The current was set at 0.1 mA cm⁻² for all the above galvanostatic measurements.

When the charging curves in Fig. 4a are examined closely, one may notice multiple voltage steps at each voltage plateaus, which is ascribed to the direct oxidation of LiOOH·H₂O and LiOH·H₂O on the cathode alongside the reactions with redox mediators in the electrolyte. To avoid the complexity and ambiguity, a RFLOB cell was fabricated, which employed ethyl viologen diiodide (EVI₂) as a bifunctional redox mediator for both the ORR and OER reactions (Fig. 4b). One significant advantage for RFLOB over the conventional Li-O₂ battery is that upon discharging O₂ is fed into a gas diffusion tank (GDT) in which it is reduced by EV⁺ when the catholyte circulates through (Supplementary Fig. 10c). As a result, the discharge product is chemically formed in the tank instead of being deposited on the electrode surface. Upon charging, the parallel reactions of these materials on the electrode are thus obviated. As the voltage profiles in Fig. 4c shows, the presence of water in the catholyte considerably reduces the cell overpotential, similar to that observed in static cells. The discharging process of the flow cells involves the reduction of EV²⁺ on the electrode and the associated ORR reaction in GDT. Upon charging, the dry cell exhibits two voltage plateaus with the first one relating to the oxidation of I⁻ to I₃⁻, and the second with extended capacity originating from the oxidation of I₃⁻ to I₂ and the reaction between Li₂O₂ and I₂, which is consistent with our previous study[17]. In comparison, although the water-contaminated flow cell also shows two clearly defined plateaus during the charging process, the capacities for both plateaus stretch nearly equally beyond that of the redox mediators, implying two distinct reactions with I₃⁻ and I₂ take place at the two plateaus respectively.

The redox-targeting reactions of LiOOH with I₃⁻ and LiOH with I₂ were further investigated with UV-Vis and mass spectrometry. After mixing with LiOOH·H₂O suspension in DME, the characteristic absorption peak of I₃⁻ at 364 nm (extended to the visible region, Supplementary Fig. 14a) vanished. As a result, the solution became nearly colourless. Meanwhile, the mass spectrometric measurement in Supplementary Fig. 15a shows that O₂ evolves instantaneously upon mixing I₃⁻ with LiOOH·H₂O. In a separate test, after adding excessive LiOH into a solution of I₂ in DME/H₂O (10:1) and stirring for 1 h, the absorption of the I₂ solution became fairly identical to that of I₃⁻ (Supplementary Fig. 14b). That is, the absorption at 400–500 nm was greatly attenuated with only the characteristic peak of I₃⁻

present, indicating the existence of I₃⁻ after reaction. We noticed IO₃⁻ was detected to be the main product of the reaction between I₂ and LiOH in water-based electrolyte in the literature[26]. The involved reaction is, 3I₂ + 6LiOH → 5LiI + LiIO₃ + 3H₂O. Considering both I⁻ and IO₃⁻ are colourless, the above reaction seems unlikely to be predominant in the DME/H₂O (10:1)-based electrolyte. The mass spectrometric measurement of the reaction between LiOH and I₂ was conducted in two different solutions. As shown in Supplementary Fig. 15b, oxygen evolution was observed instantaneously after I₂ was injected into 2 M LiOH suspension in DME/H₂O (10:1), further confirming the irrelevance of the above reaction in the DME/H₂O (10:1)-based electrolyte. Therefore, O₂ evolution is deemed part of the reactions of LiOOH with I₃⁻ and LiOH with I₂ during the charging process of water-contaminated Li-O₂ cells.

**Two-stage charging reactions in water-contaminated Li-O₂ cells.** On the basis of the above analysis, the following reactions are thus proposed to expound the discharging and the two-stage charging processes in the water-contaminated cell:

Discharging process (ORR reaction):

$$2Li^+ + O_2 + 3H_2O + 2e^- \rightarrow LiOOH \cdot H_2O + LiOH \cdot H_2O \quad (4)$$

Charging process (OER reaction):

$$3I^- \rightarrow I_3^- + 2e^- \quad (5)$$

$$4LiOOH \cdot H_2O + 2LiI_3 \rightarrow 6LiI + 3O_2 + 6H_2O \quad (6)$$

$$I_3^- \rightarrow 3/2I_2 + e^- \quad (7)$$

$$4LiOH \cdot H_2O + 6I_2 \rightarrow 4LiI_3 + O_2 + 6H_2O \quad (8)$$

Reaction (4) indicates that equimolar amounts of LiOOH and LiOH are formed in the discharging process, which is evidenced by the identical capacity extension in the two-stage charging process in Fig. 4a,c. OOH⁻ has been proposed in battery reactions in several studies as a result of H₂O dissociation or electrolyte decomposition[25,28,32,39], whereas none of them explicitly indicates LiOOH or LiOOH·H₂O as a distinct discharge product, nor its structural and electrochemical properties. Besides, we have noticed that water has substantial influence on the morphology of the discharge product. When 9.1 vol.% water was added into the electrolyte, rod and cube-like

crystals with clear edges were observed after discharge, whereas crystals formed in dry electrolyte tested in separate cells are agglomerated into round-shaped particulates (Supplementary Fig. 16). Reactions (5) and (7) describe the two-stage charging process, associated with the redox-targeting reactions with $LiOOH \cdot H_2O$ (6) and $LiOH \cdot H_2O$ (8), respectively. Reaction (6) represents a disproportionation reaction with $\frac{3}{4}$ of the peroxide being oxidized to $O_2$, and the remaining $\frac{1}{4}$ reduced to $H_2O$. Additional $\frac{1}{4}$ $O_2$ is produced in reaction (8) $via$ a 4-electron process, making up the total $O_2$ consumption in the discharging reaction (4). A consequence of such two-step reactions is that both the voltage efficiency and energy efficiency are compromised.

## Discussion

The above study unveils an intriguing picture of the ORR and OER reactions for water-contaminated $Li-O_2$ batteries, for which the transfer of protons from water leads to the formation of a new compound—$LiOOH \cdot H_2O$, while the left-over $OH^-$ prompts the formation of an equimolar amount of LiOH (or $LiOH \cdot H_2O$). More precisely, such a situation might be called water contamination at 'neutral' conditions. It would be interesting to consider the scenarios under varying proton concentrations. For instance, in weakly acidic condition, it is likely that $Li^+ + O_2 + H^+ + 2e^- + H_2O \rightarrow LiOOH \cdot H_2O$, so that $LiOOH \cdot H_2O$ would again be the discharge product with pH-dependent equilibrium potential. However, when excessive protons present in the catholyte, it appears plausible that $2H^+ + O_2 + 2e^- \rightarrow HOOH$. That is, the remaining Li atom in $LiOOH \cdot H_2O$ could be knocked out to form $H_2O_2$. At the other extreme, when extra $OH^-$ is introduced into the catholyte, the proton in LiOOH would be removed and as reported for water-based alkaline electrolytes[40], the formation of $Li_2O_2$ would be favoured, for which the redox potential of the overall reaction becomes pH-independent. Figure 5 illustrates the plausible battery reactions of 'proton-contaminated' $Li-O_2$ cells which may predominate at different $[H^+]$. We believe systematic studies on the impact of protons on the ORR and OER reactions would disclose deeper insights into the mechanistic understanding of the battery chemistry of Li-air batteries.

Considering the $LiOOH \cdot H_2O$ has greater reactivity towards the OER reaction, which substantially brings down the charging overpotential (Fig. 3b), it is of immediate importance to contemplate the implications of the new compound for the operation of the $Li-O_2$ battery. However, this seems not intuitively straightforward. A formation of $LiOOH \cdot H_2O$ during the discharging process requires protons which are not available from the anode. Although the moisture in air could be a natural source of protons, the accumulation of water in the catholyte during the charging process makes it unsustainable. In addition, the formation of $LiOOH \cdot H_2O$ in water- or alcohol-containing electrolytes may provide an alternative approach of reactions for water-based $Li-O_2$ battery, in which the 4-electron process is generally considered. As water could be precluded as a reactant from the 2-electron process of LiOOH (that is, in acidic and basic conditions), it may in theory boost the energy density and energy efficiency of water-based $Li-O_2$ cells.

A distinct battery chemistry was discovered for water-contaminated $Li-O_2$ battery, from which a new lithium compound—$LiOOH \cdot H_2O$, was identified as a predominant oxygen reduction product and structurally characterized. When iodide is used as the OER redox catalyst in the water-contaminated $Li-O_2$ cell, the equimolar amounts of $LiOOH \cdot H_2O$ and $LiOH \cdot H_2O$ formed in the discharging process are oxidized stepwise by $I_3^-$ and $I_2$, leading to a two-stage charging process.

This study discloses that the moisture fed into the cell does not pose immediate adverse impact to the battery operation, so long as the lithium anode is properly protected. On the basis of this new battery chemistry, a panoramic view of the ORR/OER reactions at different $[H^+]$ is conceived, which is anticipated to provide deeper insights into the mechanistic understanding of the chemistry of Li-air batteries. For that, we believe a more systematic study would be desired in future to understand the factors such as water content, $[H^+]$, type of redox mediators and so on, that influence the formation of $LiOOH \cdot H_2O$ or other oxygen reduction products.

## Methods

**Materials.** LiOH (98%, Sigma-Aldrich), $Li_2O_2$ (90%, Sigma-Aldrich), and $H_2O_2$ (35% (w/w) in $H_2O$, Alfa Aesar) were used in the titration experiments. LiOH (98%, Sigma-Aldrich) and $H_2O_2$ (35% (w/w) in $H_2O$, Alfa Aesar) were used for the preparation of $LiOOH \cdot H_2O$. DME(99%, Sigma-Aldrich) and dimethyl carbonate (anhydrous, $> 99\%$, Sigma-Aldrich) were used to get the precipitation of $LiOOH \cdot H_2O$. Diethylene glycol dimethyl ether (DEGDME, 99%, Sigma-Aldrich), DME (99%, Sigma-Aldrich), dimethyl sulfoxide (DMSO, 99.9%, Sigma-Aldrich) and lithium bis(trifluoromethane)sulfonimide (LiTFSI, 99.95%, Sigma-Aldrich) were used as solvent and lithium salt for electrolyte preparation. LiI (99%, Sigma-Aldrich) and $EVI_2$ (99%, Sigma-Aldrich) were used as redox mediators in $Li-O_2$ batteries. A $Li^+$-conducting ceramic membrane (LAGP, area 2 cm × 2 cm, thickness 0.5 mm) was used as separator in $Li-O_2$ batteries. Prior to use, all the above chemicals were stored in an argon-filled glove box without exposure to air.

**Preparation of $LiOOH \cdot H_2O$.** 0.196 g LiOH (98%, Sigma-Aldrich) was added into 2.6 ml deionized water. Then the solution was stirred until LiOH powder was dissolved completely. With good stirring, 1.4 ml $H_2O_2$ solution (35% w/w, Alfa Aesar) were added dropwise into the above LiOH solution over 30 min, which resulted in 2 M $LiOOH \cdot H_2O$ solution in water. In order to retrieve solid $LiOOH \cdot H_2O$ from the solution, 2 ml DME were added dropwise into 1 ml of the above $LiOOH \cdot H_2O$ solution with stirring for 10 min. During the process, a white $LiOOH \cdot H_2O$ precipitate appeared and sedimented. In the RDE and battery measurements, 2 M $LiOOH \cdot H_2O$ solution in water was employed directly to prepare suspension of $LiOOH \cdot H_2O$. For the Raman and XRD measurements, wet $LiOOH \cdot H_2O$ particles were separated from the supernatant after centrifugation and gently dried in a vacuum oven at room temperature for 12 h.

**Assembly of $Li-O_2$ battery.** Assembly of static $Li-O_2$ battery: Lithium foil and carbon felt were used as the anode and cathode, respectively. The electrochemical cell was fabricated by sandwiching the lithium foil and carbon felt in a cell stack, in which the two electrodes were separated by a LAGP membrane mounted on a Teflon frame (Supplementary Fig. 6). The effective area of the membrane was $1 \times 1 \, cm^2$. The anodic and cathodic end plates were made of stainless steel and titanium metal (with holes as $O_2$ inlet and outlet), respectively, to prevent corrosion caused by the redox species. The anodic compartment was filled with electrolyte consisting of 0.5 M LiTFSI/DEGDME and the cathodic compartment was filled with 0.5–0.7 ml of 0.5 M LiTFSI/DME with around 40–50 mM LiI and varying quantity of water. For water-containing electrolyte, water was mixed uniformly with the electrolyte before injected into the cell. After the $Li-O_2$ cell was assembled, the electrolyte was introduced into the cell, which was then tested in oxygen bag filled with pure oxygen.

Assembly of redox flow $Li-O_2$ battery: Redox flow $Li-O_2$ cell consists of a battery stack and a GDT. The cell procedure of fabricating the cell stack is the same to the static $Li-O_2$ cell. The cell stack was connected to the GDT tank by Teflon tubing, through which the catholyte was circulated between the cell and GDT tank by a peristaltic pump (Supplementary Fig. 6). The anolyte was 0.5 M LiTFSI/DEGDME. The catholyte was 15 mM $EVI_2$ in 0.5 M LiTFSI/DEGDME-DMSO (1:1 v/v) with or without 9.1 vol.% $H_2O$. DMSO was used to reduce the volatility of the catholyte. The volume of catholyte was 4 ml. Constant $O_2$ flow was provided to the GDT tank ($O_2$ pressure $\sim$1 atm) during the discharging process.

**Electrochemical measurements.** The RDE measurements were conducted by using a PINE AFMSRCE rotator. The electrolyte was 1 mM LiI in 0.1 M LiTFSI/DME with 30 mM $LiOOH \cdot H_2O$, $Li_2O_2$ or LiOH dispersed in the solution. For comparison, the same measurements were conducted in the absence of LiI or lithium compounds suspension in the electrolyte solution. Pt disc (diameter 12 mm) and Pt plate were used as working and counter electrode, respectively. And $Ag/AgNO_3$ electrode was used as reference electrode. The rotating rate was 1,200 r.p.m. and the scan rate was $0.01 \, V \, s^{-1}$.

The cyclic voltammetry measurements for 2.5 mM LiI in 0.5 M LiTFSI/DME and 2.5 mM $EVI_2$ in 0.5 M LiTFSI/DEGDME-DMSO (1:1 v/v) were conducted with a scan rate of $0.02 \, V \, s^{-1}$. The working electrode was Pt disc electrode. Both counter and reference electrodes were Li metal. Differential pulse voltammetry

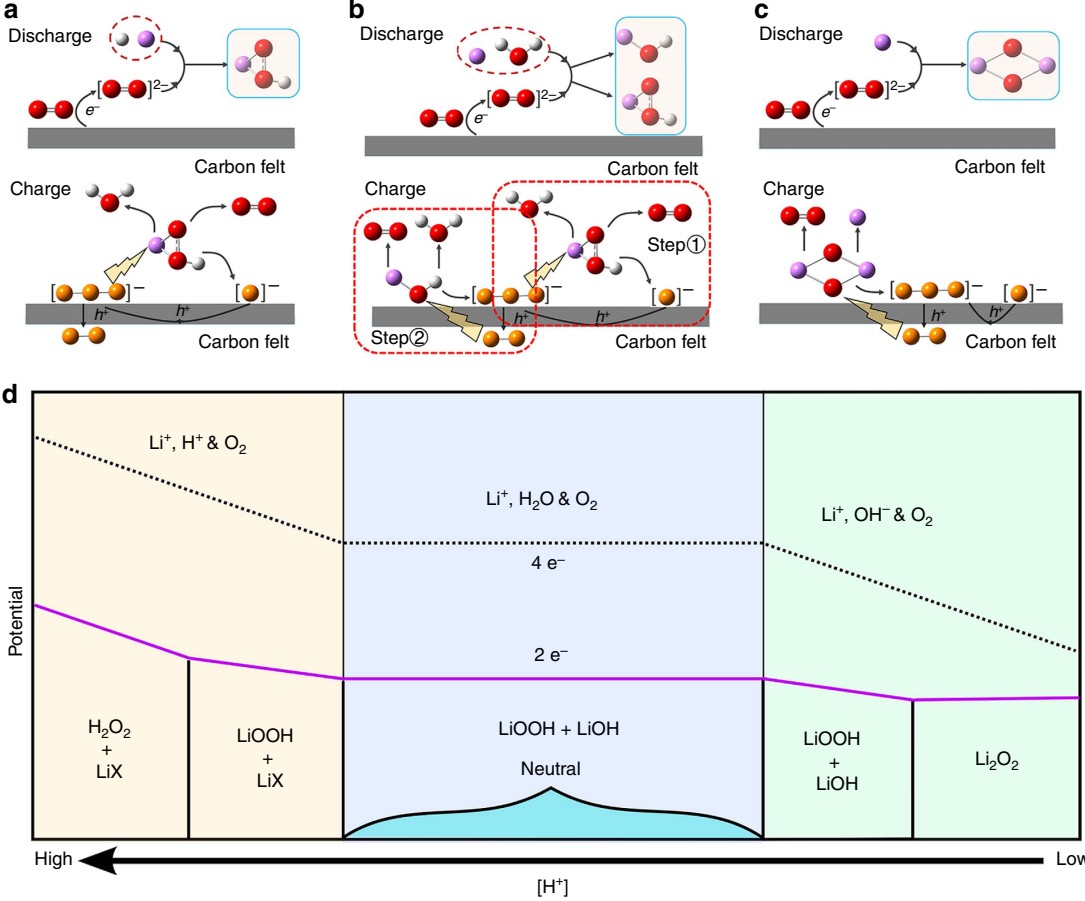

**Figure 5 | The proposed mechanism of proton-contaminated Li-O2 batteries and Pourbaix diagram with different proton concentration.** The proposed mechanisms of the charging and discharging processes in 'proton-contaminated' aprotic Li-O$_2$ battery at (**a**) acidic, (**b**) neutral and (**c**) basic conditions. Iodide is included to mediate the OER reaction. Elements in the ball-and-stick model: red-oxygen; purple-lithium; write-hydrogen; yellow-iodine. (**d**) A sketch of the Pourbaix diagram showing the predominant battery chemistries of Li-O$_2$ cell at different [H$^+$]. The 4-electron process shown in dotted line is just for reference and the potential relative to that of the two-electron process has no physical significance. HX is the acid introduced in the electrolyte.

measurements of LiI were conducted at a variety of water contents. The electrolyte was 5 mM LiI in 0.5 M LiTFSI/DME with different water contents (DME:H$_2$O are 100:0, 100:1, 100:5, 100:10, 100:20, 100:30, 100:50, 100:100, v/v). The working and the counter electrode were Pt disc and Pt plate electrode, respectively. The reference electrode was Ag/AgNO$_3$ electrode. The step potential is 0.005 V and the modulation amplitude is 0.025 V. All the above measurements were performed on an Autolab electrochemical workstation (Metrohm Autolab, PGSTAT302N). The charge and discharge tests were conducted on an Arbin battery tester. The battery was tested at a constant current of 0.1 mA cm$^{-2}$.

**Other characterizations.** Raman spectra were measured by a confocal Raman system with 532 nm laser excitation (WITec Instruments Corp, Germany). Samples for Raman spectroscopic measurement include gently dried LiOOH · H$_2$O powder, powders of the as-purchased LiOH and Li$_2$O$_2$, as well as 35 % H$_2$O$_2$ solution. The Raman spectrum of the cathode (carbon felt) in the Li-O$_2$ batteries was also measured immediately after disassembling the fully discharged static Li-O$_2$ cells. For the Li-O$_2$ cells with water-free electrolyte, the cathode was washed and dried in a vacuum chamber prior to Raman measurement. All the above samples were loaded on a piece of sapphire for Raman measurement. FTIR spectra were collected *via* PerkinElmer Frontier MIR/FIR system by 16 scans with a nominal resolution of 1 cm$^{-1}$ through an ATR mode. In the *in situ* measurement of the redox-targeting reaction product between EV$^+$ and O$_2$, a flow cell with two inlets and one outlet (two for liquid and one for oxygen) was used. UV-Vis spectroscopic measurements were conducted on a UV-Vis spectrophotometer (Shimadzu 1800). Mass spectrometric measurements were conducted on a Hiden analytical QGA (HAS-301-1376A). There are one outlet and two inlets of the reactor for carrying gas and injecting reactant.

The high-energy XRD measurements were performed on the beam line 11-ID-C at the Advanced Photon Source, Argonne National Laboratory. A monochromator with a Si (113) single crystal was used to provide an X-ray beam with the energy of 115 keV. High-energy X-ray with a beam size of 0.2 mm × 0.2 mm and wavelength of 0.10725 Å was used to obtain two-dimensional diffraction patterns in the transmission geometry. X-rays were collected with a Perkin-Elmer large-area detector placed at 1,800 mm from the sample. The obtained two-dimensional diffraction patterns were calibrated using a standard CeO$_2$ sample and converted to one-dimensional patterns using Fit2D software. In order to figure out the structure of the obtained LiOOH · H$_2$O sample, the high-resolution X-ray powder diffraction pattern of LiOOH · H$_2$O was taken at 11-BM, Advanced Photon Source, Argonne National Lab, whereas the wavelength of the X-ray is 0.41423 Å. The samples were measured in air without protection. For the stability test, XRD measurements were carried out on a Bruker D8 with Cu Kα1 radiation ($\lambda = 0.154059$ nm). The samples were measured in air without protection.

**Theoretical calculations.** DFT calculations were carried out with B3LYP hybrid exchange-correlation functional in combination with the quadruple-zeta polarized valence basis set augmented with diffuse functions, aug-cc-pVQZ, using the Gaussian 09 programme suite[41,42]. The scaled quantum mechanics force field procedure was used to analyse vibrational bands of all fundamentals. The calculated frequencies were scaled by a factor of 0.9852 for frequencies below 2,000 cm$^{-1}$ (ref. 43). Scaling harmonic vibrational frequencies is an effective way to facilitate comparison with experimentally observed frequencies. A scaling factor of 0.9852 was recommended for the B3LYP/aug-cc-pVQZ level of theory for which the corresponding root mean square error relative to the experimental frequencies was report to be 8 cm$^{-1}$ (ref. 43). The calculated scaled frequencies for the O–O stretch are: 838.0 (LiOOH · H$_2$O), 827.6 (LiOOLi), 934.9 (HOOH) cm$^{-1}$. The structure and vibration frequencies of LiOOH · H$_2$O are also verified against partial phonon density of states generated using Phonopy package with Vienna ab initio simulation package and GGA-PBE exchange and correlation potential.

**Data availability.** The authors declare that data supporting the findings of this study are available within the paper and its supplementary information file or from the corresponding author on reasonable request.

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

## Acknowledgements

This research was supported by the National Research Foundation, Prime Minister's Office, Singapore, under its Competitive Research Program (CRP Awards No NRF-CRP8-2011-04 and NRF-CRP10-2012-06). We thank Dr Du Yuan for his assistance on the FTIR measurement. We thank Professor Hui Ying Yang and Dr Linfeng Sun from Singapore University of Technology and Design for their assistance on Raman measurement.

## Author contributions

Q.W. conceived the study and supervised the work. Y.G.Z. designed and performed most of the experiments. Q.L., Y. Rong and Y. Ren conducted synchrotron XRD measurements and data analysis. S.A. solved the crystal structure of LiOOH·H$_2$O. C.J. provided some scientific suggestion. X.X. prepared the LAGP membrane. J.Y., L.-J.Y., A.K. and H.C. did the DFT calculations. Y.G.Z., S.A. and Q.W. wrote the paper.

## Additional information

**Competing financial interests**: The authors declare no competing financial interests.

**How to cite this article**: Zhu, Y. G. *et al.* Proton enhanced dynamic battery chemistry for aprotic lithium–oxygen batteries. *Nat. Commun.* **8,** 14308 doi: 10.1038/ncomms14308 (2017).

**Publisher's note**: 

