## [Peer Review File · Nature Communications]

Reviewers' comments:

Reviewer #1 (Remarks to the Author):

This paper introduces LiOOH·H₂O as a major discharge product in lithium-oxygen battery. This compound may greatly influence the discharge/charge behavior, but has never been reported before. The conclusions of this paper are based on solid evidences and are very helpful for battery builders to correctly understand the lithium-oxygen chemistry. It is generally an inspirational paper. But some revisions are needed to improve the manuscript:

1. Line 174, Figure 2(b) should be Figure 3(b).
2. Line 84, "The reactivity of the three lithium compounds towards oxidation by I₃⁻ is determined in the sequence: LiOOH < I₃⁻ < Li₂O₂ < LiOH." Why is I₃⁻ itself in the sequence? The LiOOH should have the highest reactivity with I₃⁻.
3. The authors claimed "At 0 K the LiOOH·H₂O should according to the DFT calculations be marginally stable against the decomposition into LiOH·H₂O and ½ O₂." Is LiOOH·H₂O stable at room temperature? Does the XRD pattern change with time? Please clarify this point in the manuscript.
4. The Raman spectroscopy of the discharge product in anhydrous electrolyte should be exhibited in Figure 1a.
5. The reaction between LiOH and I₂ (reaction 6') should be confirmed with titration experiment as Figure S1. And I am not sure if "titration" is a proper word.
6. The authors mentioned that LiOOH·H₂O can easily decomposed into Li₂CO₃ when soaked in the DMC, the specific mechanism should be explained.
7. In Figure 5b, it would be helpful if the authors can give a quantitative description of the proton concentration.

Reviewer #2 (Remarks to the Author):

Remarks to the Author:

Comments on the paper NCOMMS-16-19441 titled "Proton Enhanced Dynamic Battery Chemistry for Aprotic Li-O₂ Batteries"

Proton or water containing aprotic Li-O₂ battery generated widespread controversy due to the diverse discharge products (Li₂O₂, LiOH) and ambiguous reaction mechanism. Hence, clearly illustrating the impact of proton and/or proposing new battery reaction mechanism is significant for Li-O₂ field. Interestingly, LiOOH·H₂O is identified as an unprecedented discharge product in water-doped Li-O₂ cells, moreover, this newly-found product is prone to react with redox catalyst (I₃⁻) leading to low charging overpotential in this work. The experiment is well designed and the manuscript is well written, and may have strong impact on Li-O₂ arena concerning proton/water contaminated or soluble catalyst-based Li-O₂ cells. This manuscript may be acceptable after addressing the following technical issues. The Raman spectra (Fig 1a) is suspicious, the background signal of glass substrate is extremely strong, which leads to the distortion of characteristic peak of Li-O in LiOH. So the reviewer strongly suggests that the Raman test should be repeated using quartz or sapphire to eliminate the strong peak of substrate. Moreover, according to the reaction mechanism (4), LiOOH·H₂O and LiOH·H₂O are co-existed as discharge products after discharge, however, no clear evidences for the existence of LiOH·H₂O can be found in Fig 1a, more tests such as XRD, FTIR for discharged electrode should be done to verify the proposed discharge mechanism (4).

In order to ascertain the claimed mechanism for charging procedure (5)(5')-(6)(6'), it is necessary to provide direct evidences that various reaction occurs at different recharge stages as illustrated in

Fig4a,c. The additional XRD and/or Raman experiments on the air electrodes at each stage of charge, which can intuitively confirm the oxidation of deposited species, are suggested to carry out to further reveal the reaction process.

Moreover, from the proposed reaction mechanism, gaseous product O₂ is regarded as important indicator for the hypothetical battery reaction (4)-(6). Dynamic quantifying the amount of gaseous products by DEMS, which has been widely explored by Bruce, McCloskey et al is indispensable to interrogate the reaction mechanism and distinguish discharge products. The reviewer strongly suggests performing DEMS tests for dis-recharge processes.

The titration experiments (Fig S1) illustrate that the reaction rate of seems fast, besides, the electrochemical reaction of I⁻/ I₃⁻ during the charge (state I) is instantaneous. Once the discharged electrode has already deposited by LiOOH·H₂O, the reaction (LiOOH·H₂O and I₃⁻) may instantaneously occur at early stage of recharge. However, the authors demonstrate long charging time of I⁻ to I₃⁻ (Fig 4a,c), after that, occurs the reaction between LiOOH·H₂O and I₃⁻. But the review speculates that two reactions (I⁻ to I₃⁻/ LiOOH·H₂O and I₃⁻) may simultaneously happen and estimated deposited LiOOH·H₂O with little amount can be totally eliminated at this initial recharge stage. The following oxidation of LiOOH·H₂O shown in the Figure 4a,c seems irrational. Please authors make comments on this issue.

Some clerical errors should be revised, for example:

Line 99, Figure S7, the authors should cite the Figures in sequence and the authors should readjust the order of figures in SI.

Line 127, "cf. Fig.2c" what does the "cf" mean?

Line 174, "Figure 2(b)" is wrong.

Reviewer #3 (Remarks to the Author):

"The authors suggest that they have identified a new phase, LiOOH hydrate, after discharging a water contaminated non-aqueous Li-O₂ battery. Moreover, this new phase appears to be more rechargeable using the iodide mediator. These observations are very interesting and can potentially be published in Nature Communications. However, further experiments and reviewing are needed.

First of all, the manuscript is poorly written, with many grammar mistakes and typos, making the text difficult to follow. Please remove subjective phrases that appear at many places in the manuscript, such as "not surprisingly", "strikingly", and be neutral of the results. I suggest that the edited manuscript be more seriously proofread before further submission.

The scientific points need to be addressed at this stage:

(1) The authors claim that LiOOH·H₂O is the predominant discharge product (abstract) in a water added non-aqueous Li-O₂ battery, but there is insufficient evidence to support this statement — throughout the manuscript there is only one noisy Raman spectrum (fig. 1(a)) suggesting so; the rest (Raman, IR, XRD) are based on the synthesized model compound. To demonstrate the relevance of LiOOH·H₂O to a reversible Li-O₂ batteries, the authors need to show high quality spectroscopic data, e.g., Raman, IR etc, at the end of discharge and charge. In line 254, the authors suggest in reaction

(4) equal moles of LiOOH-H₂O and LiOH-H₂O would form during discharge, contradicting to the statement in the abstract that LiOOH-H₂O is the predominant discharge product. Therefore, the authors should provide rationales and experimental evidence to support either the absence or the existence of LiOH-H₂O in the discharge product.

(2) The authors suggest LiOOH-H₂O can be decomposed by I₂ and LiOH-H₂O can be decomposed by I₂ to evolve O₂. It is necessary to perform mass spectrometry to confirm this. Some quantification (IR, Raman, titration etc.) of LiOH and LiOOH removal are also required in the experiment described in Fig. 3(b), rather than speculating the reaction between hydroxide/peroxide phases and iodine species entirely based on capacities of electrochemistry.

(3) The authors need to show more results (IR, Raman, XRD etc) and discuss further on the role of water on the battery chemistry: at what specific water content does the LiOOH-H₂O form in the discharge battery; is there a range of water content, where LiOOH-H₂O forms or not, and why; showing this is essential so that the result can be reproducible. Please state in detail the water content in terms of wt%, vol% and articulate how this water added electrolyte is prepared."

REVIEWERS' COMMENTS:

Reviewer #1 (Remarks to the Author):

The revised manuscript is improved a lot. All of the referee's questions are well answered. It is suitable for publication.

Reviewer #2 (Remarks to the Author):

Comments on the revised paper titled " Proton Enhanced Dynamic Battery Chemistry for Aprotic Li-O₂ Batteries "

This revised manuscript " Proton Enhanced Dynamic Battery Chemistry for Aprotic Li-O₂ Batteries " revised by Wang and coworkers, clearly illustrates a distinct battery chemistry mechanism in water-contaminated aprotic Li-O₂ batteries and LiOOH·H₂O is identified as an unprecedented discharge product, providing a new insights into the understanding of Li-O₂ cell. Moreover, the added or revised experimental results and conclusions are complete and compelling in this revision form, the authors meticulously and accurately reply to the questions which suggested by the reviewer. So the reviewer strongly recommends that this revised manuscript has a qualification for publishing in Nat. Comm. without further revision.

Reviewer #3 (Remarks to the Author):

In the revised manuscript, much more evidence has been shown to support the interesting observation reported. I therefore recommend for publication. Nice work.

Replies to the comments of Reviewer #1:

This paper introduces LiOOH·H₂O as a major discharge product in lithium-oxygen battery. This compound may greatly influence the discharge/charge behavior, but has never been reported before. The conclusions of this paper are based on solid evidences and are very helpful for battery builders to correctly understand the lithium-oxygen chemistry. It is generally an inspirational paper. But some revisions are needed to improve the manuscript:

Reply: We thank the reviewer for the positive and constructive comments. We have revised the manuscripts based on the reviewer's comments and suggestions.

1. Line 174, Figure 2(b) should be Figure 3(b).

Reply: Thanks for pointing out the error. We have corrected the typo.

2. Line 84, "The reactivity of the three lithium compounds towards oxidation by I₃⁻ is determined in the sequence: LiOOH < I₃⁻ < Li₂O₂ < LiOH." Why is I₃⁻ itself in the sequence? The LiOOH should have the highest reactivity with I₃⁻.

Reply: The reason we included I₃⁻ in the comparison is to highlight that I₃⁻ is able to oxidize LiOOH, but not Li₂O₂ at the time scale of titration experiment. The latter has been proved — although thermodynamically the potential of I₃⁻ is higher than Li₂O₂, the sluggish kinetics make the reaction invisibly slow. To avoid confusion, we have changed the statement to “on the basis of the above titration tests, the reactivity of LiOOH is the highest and LiOH is the lowest towards oxidation by I₃⁻”.

3. The authors claimed "At 0 K the LiOOH·H₂O should according to the DFT calculations be marginally stable against the decomposition into LiOH·H₂O and ½ O₂." Is LiOOH·H₂O stable at room temperature? Does the XRD pattern change with time? Please clarify this point in the manuscript.

Reply: We have conducted powder XRD and Raman spectroscopic measurements of LiOOH·H₂O to monitor the stability of the material when exposed to air for different durations. The results have been included in the supporting information.

As shown in Figure R1, the stability of LiOOH·H₂O was measured by using XRD measurements. Firstly, we tested the wet powder of LiOOH·H₂O. Then, we tested the powder left in air for 10 min and 2 hours. The formations of LiOH·H₂O, LiOH, and Li₂CO₃ are clearly seen as time evolves, of which Li₂CO₃ is presumably a reaction product of LiOOH·H₂O or LiOH with CO₂ from air. In order to rule out the influence of CO₂, we dried LiOOH·H₂O in vacuum condition and did another XRD measurement, for which the main products become LiOH and LiOH·H₂O, validating the above speculation.

Figure R1. XRD patterns of LiOOH·H₂O after exposed in air or vacuum conditions.

In a separate test in Figure R2, the Raman spectra of LiOOH·H₂O evolved gradually and the characteristic peaks of LiOH and Li₂CO₃ become obvious even after 5 min exposure in air.^{1,2} The LiOOH was entirely converted into LiOH and Li₂CO₃ after 45 min, indicating LiOOH·H₂O is not stable in air. This result is consistent with the XRD measurement.

Figure R2. Raman spectra of LiOOH·H₂O exposed in air for different durations. The LiOOH·H₂O was initially in the form of wet powder.

4. The Raman spectroscopy of the discharge product in unhydrous electrolyte should be exhibited in Figure 1a.

Reply: The Raman spectrum of the discharge product in water-free electrolyte has now been included in Figure 1b. In the new measurement, we replaced the soda lime glass with sapphire to eliminate the signals from the substrate. As shown in Figure R3, Li_2O_2 as the main discharge product is clearly seen after fully discharging a static cell with water-free electrolyte. This is evidentially distinct from that obtained in water-containing electrolyte.

Figure R3. Raman spectra of the three lithium compounds ($\text{LiOOH}\cdot\text{H}_2\text{O}$, LiOH and Li_2O_2). The Raman spectra of the discharge products with or without water in the electrolyte are also exhibited.

5. The reaction between LiOH and I_2 (reaction 6') should be confirmed with titration experiment as Figure S1. And I am not sure if "titration" is a proper word.

Reply: As suggested by the reviewer, the titration experiment of LiOH in I_2 solution has been performed. After adding excessive LiOH and stirring for 1 hour, the colour of I_2 solution in $\text{DME}/\text{H}_2\text{O}$ (10:1) became slightly lighter (Figure R4a). UV-Vis measurement was conducted to monitor the colour change. As shown in Figure R4b, after reacting with LiOH , the absorption of the I_2 solution became identical to that of the standard solution of I_3^- . That is, the absorption at 400-500 nm was greatly attenuated with only the characteristic peak of I_3^- presented at around 364 nm, which confirms the existence of I_3^- in the reacted I_2 solution. The absorption peak broadening of I_2 solution after titration may be attributed to the scattering effect of small particles of LiOH , which has limited solubility in $\text{DME}/\text{H}_2\text{O}$ (10:1). In addition, considering both I^- and IO_3^- are colourless, $3\text{I}_2 + 6\text{LiOH} \rightarrow 5\text{LiI} + \text{LiIO}_3 + 3\text{H}_2\text{O}$ is unlikely to be the main reaction in $\text{DME}/\text{H}_2\text{O}$ (10:1) solution. It is more likely that $6\text{I}_2 + 4\text{LiOH} \rightarrow 4\text{LiI}_3 + \text{O}_2 + 2\text{H}_2\text{O}$ (reaction 6'). This is further corroborated with mass spectrometric measurement.

Figure R4. (a) Pictures of 10 mM I_2 solution show the color change after different treatments with LiOH. (b) The UV-Vis spectra of standard 0.1 mM I_3^- solution and the reacted I_2 solution with LiOH. The inset of (b) shows the photos of 6.6 mM I_3^- (1), 10 mM reacted I_2 (2, DME / H_2O = 10 / 1), 10 mM I_2 (3), and 10 mM reacted I_2 (4, DME / H_2O = 1 / 1) solutions.

The mass spectrometric measurement of the reaction between LiOH and I_2 was conducted in two different solutions. As shown in Figure R5, oxygen evolution was observed immediately after I_2 was injected into the 2 M LiOH suspension in DME/ H_2O (10:1). Therefore, O_2 evolution is deemed part of the reaction between I_2 and LiOH in the DME/ H_2O (10:1) electrolyte system, which is the same electrolyte used in the Li- O_2 cell. In contrast, no oxygen was detected when the solvent was changed to DME/ H_2O (1:1). We noticed IO_3^- was detected to be the main product of the reaction between I_2 and LiOH in water-based electrolyte in the literature.³

As this reaction does not involve O_2 evolution, those happened in Li- O_2 battery apparently follows a different route as that indicated in reaction 6'. And the concentration of water is a key factor dictating the reaction between I_2 and LiOH.

Figure R5. Oxygen evolution recorded from the reaction between I_2 and LiOH monitored by mass spectrometry. The reaction was performed in solutions with different concentration of H_2O . The starting solutions in the reactor are 2 mL 2 M LiOH in DME/ H_2O (10:1) and 2 mL 2 M LiOH in H_2O .

6. The authors mentioned that $LiOOH \cdot H_2O$ can be easily decomposed into Li_2CO_3 when soaked in the DMC, the specific mechanism should be explained.

Reply: It is proposed in the aprotic electrolyte system that the intermediate $O_2^{\cdot -}$ promotes the decomposition of carbonate solvent during the discharging process.⁴⁻⁶ In our study, $LiOOH \cdot H_2O$ was found to be more reactive than Li_2O_2 , which could plausibly be a reason for the formation of Li_2CO_3 in DMC. However, the decomposition mechanism of DMC in the presence of $LiOOH \cdot H_2O$ is yet clear although we believe OOH^- may play a critical role. Considering DMC was not employed as the electrolyte solvent in this work, we decide to remove this part from the supporting information to avoid confusion. We will systematically study the reactivity of $LiOOH \cdot H_2O$ towards various aprotic electrolytes in future.

7. In Figure 5b, it would be helpful if the authors can give a quantitative description of the proton concentration.

Reply: While we fully agree with the reviewer that a quantitative description of the proton concentration would be interesting for the Pourbaix diagram in Figure 5b, we found itself very difficult to reliably monitor the proton concentration in aprotic solvent, as it is highly dependent on the properties (i.e. dielectric constant, etc.) of the organic solvent (or its mixture with water) and the resulted Gibbs free energy for proton solvation. So it is hardly possible to make a universal picture quantitatively describing the influence of protons. Instead, we show the tendency when proton concentration varies, to disclose the various reaction routes for water-contaminated aprotic electrolyte.

Replies to the comments of Reviewer #2:

Remarks to the Author:

Comments on the paper NCOMMS-16-19441 titled “Proton Enhanced Dynamic Battery Chemistry for Aprotic Li-O₂ Batteries”

Proton or water containing aprotic Li-O₂ battery generated widespread controversy due to the diverse discharge products (Li₂O₂, LiOH) and ambiguous reaction mechanism. Hence, clearly illustrating the impact of proton and/or proposing new battery reaction mechanism is significant for Li-O₂ field. Interestingly, LiOOH·H₂O is identified as an unprecedented discharge product in water-doped Li-O₂ cells, moreover, this newly-found product is prone to react with redox catalyst (I³⁻) leading to low charging overpotential in this work. The experiment is well designed and the manuscript is well written, and may have strong impact on Li-O₂ arena concerning proton/water contaminated or soluble catalyst-based Li-O₂ cells. This manuscript may be acceptable after addressing the following technical issues.

Reply: We thank the reviewer for the constructive comments and encouragement.

The Raman spectra (Fig 1a) is suspicious, the background signal of glass substrate is extremely strong, which leads to the distortion of characteristic peak of Li-O in LiOH. So the reviewer strongly suggests that the Raman test should be repeated using quartz or sapphire to eliminate the strong peak of substrate. Moreover, according to the reaction mechanism (4), LiOOH·H₂O and LiOH·H₂O are co-existed as discharge products after discharge, however, no clear evidences for the existence of LiOH·H₂O can be found in Fig 1a, more tests such as XRD, FTIR for discharged electrode should be done to verify the proposed discharge mechanism (4).

Reply: Following the reviewer’s suggestion, we have repeated the Raman spectroscopic measurement of the discharge product. In the new measurement, the glass substrate was replaced with sapphire. As shown in Figure R6, the spectrum is improved, from which the characteristic peaks of LiOOH·H₂O and LiOH are clearly identified. However, the signal is still relatively noisy due to the attenuation by the sapphire cover. In addition, the electrolyte remained in the discharge product may also distract the laser beam and the signal recorded. In comparison with the reference compounds, it is confident to assign the broad peaks at ~100 cm⁻¹ and 850 cm⁻¹ to LiOH and LiOOH·H₂O, both of which have characteristic vibrations overlapping in this region. The unknown peaks in the range of 250-360 cm⁻¹ are presumably from the substrate or electrolyte.

To substantiate the presence of LiOOH and LiOH in the discharge (ORR) product, ATR-FTIR measurements were conducted. Here the ORR reaction was promoted by EV⁺, which reduces O₂ in the presence of Li⁺ and forms LiOOH and LiOH. This is the same discharging process in the redox flow lithium oxygen battery (RFLOB). As such, we firstly obtained EV⁺ electrolyte by reducing 0.1 M EV²⁺ in 0.5 M LiTFSI / (DMSO:DME, 1:1) electrolyte with Li metal. 9.1 vol.% water was then added into the EV⁺ electrolyte which was subsequently injected into a flow cell holder in O₂ atmosphere for ATR-FTIR measurement. As shown in Figure R7, the characteristic peaks of LiOH (910-1070 cm⁻¹) and LiOOH·H₂O (1641 cm⁻¹) appeared after 5 min reaction and became more pronounced after 10 min. This further verifies the formation of LiOH and LiOOH from the ORR reaction in the presence of 9.1 vol.% water in the electrolyte.

Figure R6. Raman spectra of the three lithium compounds (LiOOH·H₂O, LiOH and Li₂O₂). The Raman spectra of the discharge products with or without water in the electrolyte are also exhibited.

Figure R7. ATR-FTIR spectra of the electrolyte and the ORR reaction products by EV⁺ in the presence of 9.1 vol.% H₂O in the electrolyte.

In order to ascertain the claimed mechanism for charging procedure (5)(5')-(6)(6'), it is necessary to provide direct evidences that various reaction occurs at different recharge stages as illustrated in Fig4a,c. The additional XRD and/or Raman experiments on the air electrodes

at each stage of charge, which can intuitively confirm the oxidation of deposited species, are suggested to carry out to further reveal the reaction process.

Moreover, from the proposed reaction mechanism, gaseous product O₂ is regarded as important indicator for the hypothetical battery reaction (4)-(6). Dynamic quantifying the amount of gaseous products by DEMS, which has been widely explored by Bruce, McCloskey et al is indispensable to interrogate the reaction mechanism and distinguish discharge products. The reviewer strongly suggests performing DEMS tests for dis-charge processes.

Reply: Thanks for the suggestions. While we are able to confirm the formation of LiOOH and LiOH during the discharging process, we failed to conduct *in-situ* FTIR and Raman spectroscopic measurements to monitor the reactions during the charging process. The reason is that the characteristic vibrations of LiOOH unfortunately overlap with that of I₂ in FTIR, while the signals of Raman measurement were too weak to extract meaningful result (see the best result in Figure R6). Based on the fact that both LiOOH and LiOH present as the discharge product, we thus employed *ex-situ* UV-Vis and mass spectroscopy to sequentially monitor the reactions of LiOOH·H₂O with I₃⁻ (reaction 5') and LiOH with I₂ (reaction 6'), which are the two reactions proposed for the charging process.

Figure R8. (a) UV-Vis spectra of I₃⁻ in DME / H₂O (10 / 1) before and after adding excessive LiOOH. (b) UV-Vis spectra of standard 0.1 mM I₃⁻ solution and the reacted I₂ solution with LiOH. The inset of (b) shows the photos of 6.6 mM I₃⁻ (1), 10 mM reacted I₂ (2, DME / H₂O = 10 / 1), 10 mM I₂ (3), and 10 mM reacted I₂ (4, DME / H₂O = 1 / 1) solutions.

We firstly investigated the reaction between I₃⁻ and LiOOH·H₂O. As the UV-Vis spectra shown in Figure R8a, the characteristic absorption peak of I₃⁻ at 364 nm (extended to the visible region) vanished after mixing with LiOOH·H₂O. As a result, the solution became nearly colourless. Meanwhile, the mass spectroscopic measurement in Figure R9a shows that O₂ evolves instantaneously upon mixing I₃⁻ with LiOOH·H₂O. These observations qualitatively support reaction 5' as indicated below:

The reaction between LiOH and I₂ was also investigated with UV-Vis and mass spectroscopic measurements. After adding excessive LiOH and stirring for 1 hour, the colour of I₂ solution in DME/H₂O (10:1) became slightly lighter (inset of Figure R8b). As the UV-Vis spectra shown in Figure R8b, after reacting with LiOH, the absorption of the I₂ solution became fairly identical to that of the standard solution of I₃⁻. That is, the absorption at 400-500 nm was greatly attenuated with only the characteristic peak of I₃⁻ presented at around 364 nm, which confirms the existence of I₃⁻ in the reacted I₂ solution. The absorption peak broadening of I₂ solution after titration is presumably attributed to the scattering effect of small particles of LiOH, which has limited solubility in DME/H₂O (10:1). In addition, considering both I⁻ and IO₃⁻ are colourless, $3I_2 + 6LiOH \rightarrow 5LiI + LiIO_3 + 3H_2O$ is unlikely to be the main reaction in DME:H₂O (10:1) solution. So it is more likely that

This is further corroborated with mass spectrometry.

Figure R9. Oxygen evolution recorded from the reactions of (a) LiOOH with I₃⁻ and (b) LiOH with I₂ monitored by mass spectrometry. The reactions of LiOH with I₂ were performed in solutions with different concentration of H₂O as labelled in the figure.

The mass spectrometric measurement of the reaction between LiOH and I₂ was conducted in two different solutions. As shown in Figure R9b, oxygen evolution was observed immediately after I₂ was injected into the 2 M LiOH suspension in DME/H₂O (10:1). Therefore, O₂ evolution is part of the reaction between I₂ and LiOH in the DME/H₂O (10:1) electrolyte system, which is the same electrolyte used in the Li-O₂ cell. In contrast, no oxygen was detected when the solvent was changed to DME / H₂O (1:1). We noticed IO₃⁻ was detected to be the main product of the reaction between I₂ and LiOH in water-based electrolyte in the literature³.

This reaction does not involve O₂ evolution, our observation is thus consistent with the above reaction. Apparently, the concentration of water is a key factor dictating the reaction between I₂ and LiOH.

While the UV-Vis and mass spectrometric measurements have qualitatively corroborated the two-stage reaction mechanism proposed for the charging process, quantitative and *in-situ*

measurement will be conducted in future when the equipment such as DEMS become available.

The titration experiments (Fig S1) illustrate that the reaction rate of seems fast, besides, the electrochemical reaction of I-/ I3- during the charge (state I) is instantaneous. Once the discharged electrode has already deposited by LiOOH·H2O, the reaction (LiOOH·H2O and I3-) may instantaneously occur at early stage of recharge. However, the authors demonstrate long charging time of I- to I3- (Fig 4a,c), after that, occurs the reaction between LiOOH·H2O and I3-. But the review speculates that two reactions (I- to I3-/ LiOOH·H2O and I3-) may simultaneously happen and estimated deposited LiOOH·H2O with little amount can be totally eliminated at this initial recharge stage. The following oxidation of LiOOH·H2O shown in the Figure 4a,c seems irrational. Please authors make comments on this issue.

Reply: In Figure 4a and 4c, the labels were not meant to indicate the different stage of reactions with chronically separate electrochemical reactions of redox mediators and chemical reactions with the lithium compounds, but to quantitatively show the capacities from the reactions of redox mediators and those with lithium compounds.

We fully agree with the reviewer that at different stage of the charging process, the electrochemical oxidations of redox mediators on the electrode are instantaneously associated with the chemical reactions of the redox mediators with lithium compounds (I₃⁻ with LiOOH and I₂ with LiOH). To avoid confusion, we have modified the labelling.

Some clerical errors should be revised, for example:

Line 99, Figure S7, the authors should cite the Figures in sequence and the authors should readjust the order of figures in SI.

Reply: We thank the reviewer for the suggestion. We have rearranged the figures in supporting information.

Line 127, “cf. Fig.2c” what does the “cf” mean?

Reply: “cf.” is used in writing to introduce something that should be considered in connection with the subject you are discussing. We have changed it to “see Fig. 2c”.

Line 174, “Figure 2(b)” is wrong.

Reply: Thanks for pointing out the error. We have corrected it to “Figure 3(b)”.

Replies to the comments of Reviewer #3:

“The authors suggest that they have identified a new phase, LiOOH hydrate, after discharging a water contaminated non-aqueous Li-O₂ battery. Moreover, this new phase appears to be more rechargeable using the iodide mediator. These observations are very interesting and can potentially be published in Nature Communications. However, further experiments and reviewing are needed.

First of all, the manuscript is poorly written, with many grammar mistakes and typos, making the text difficult to follow. Please remove subjective phrases that appear at many places in the manuscript, such as "not surprisingly", "strikingly", and be neutral of the results. I suggest that the edited manuscript be more seriously proofread before further submission.

Reply: We thank the reviewer for the constructive comments. The whole manuscript has been revised carefully.

The scientific points need to be addressed at this stage:

(1) The authors claim that LiOOH-H₂O is the predominant discharge product (abstract) in a water added non-aqueous Li-O₂ battery, but there is insufficient evidence to support this statement — throughout the manuscript there is only one noisy Raman spectrum (fig. 1(a)) suggesting so; the rest (Raman, IR, XRD) are based on the synthesized model compound. To demonstrate the relevance of LiOOH-H₂O to a reversible Li-O₂ batteries, the authors need to show high quality spectroscopic data, e.g., Raman, IR etc, at the end of discharge and charge. In line 254, the authors suggest in reaction (4) equal moles of LiOOH-H₂O and LiOH-H₂O would form during discharge, contradicting to the statement in the abstract that LiOOH-H₂O is the predominant discharge product. Therefore, the authors should provide rationales and experimental evidence to support either the absence or the existence of LiOH-H₂O in the discharge product.

Reply: Following the reviewer’s suggestion, we have repeated the Raman spectroscopic measurement of the discharge product. In the new measurement, the glass substrate was replaced with sapphire. As shown in Figure R10, the spectrum is improved, from which the characteristic peaks of LiOH, LiOOH·H₂O are clearly identified. However, the signal is still relatively noisy due to the attenuation effect of sapphire cover. In addition, the electrolyte remained in the discharge product distracts the laser beam and the signal recorded. In comparison with the reference compounds, it is confident to assign the broad peaks at ~100 cm⁻¹ and 850 cm⁻¹ to LiOH and LiOOH·H₂O, both of which have characteristic vibrations and overlap in this region. The unknown peaks in the range of 250-360 cm⁻¹ are presumably from the substrate or electrolyte.

Figure R10. Raman spectra of the three lithium compounds (LiOOH·H₂O, LiOH and Li₂O₂). The Raman spectra of the discharge products with or without water in the electrolyte are also exhibited.

To substantiate the presence of LiOH and LiOOH in the discharge (ORR) product, ATR-FTIR measurements were conducted. Here the ORR reaction was promoted by EV⁺, which reduces O₂ in the presence of Li⁺ and forms LiOH and LiOOH. This is the same discharging process in the redox flow lithium oxygen battery (RFLOB). As such, we firstly obtained EV⁺ electrolyte by reducing 0.1 M EV²⁺ in 0.5 M LiTFSI / (DMSO:DME, 1:1) electrolyte with Li metal. 9.1 vol.% water was then added into the EV⁺ electrolyte which was subsequently injected into a flow cell holder in O₂ atmosphere for ATR-FTIR measurement. As shown in Figure R11, the characteristic peaks of LiOH (910-1070 cm⁻¹) and LiOOH·H₂O (1641 cm⁻¹) appeared after 5 min reaction and became more pronounced after 10 min. This further verifies the formation of LiOH and LiOOH from the ORR reaction in the presence of 9.1 vol.% water in the electrolyte.

In addition, we have revised the abstract on the description of discharge product – “Both lithium ions and protons were found to be involved in the oxygen reduction (ORR) and evolution reactions (OER), and LiOOH·H₂O and LiOH were identified as predominant materials in the discharge product.”

Figure R11. ATR-FTIR spectra of the electrolyte and the ORR reaction products by EV^+ in the presence of 9.1 vol% H_2O in the electrolyte.

(2) The authors suggest $\text{LiOOH}\cdot\text{H}_2\text{O}$ can be decomposed by I_2 and $\text{LiOH}\cdot\text{H}_2\text{O}$ can be decomposed by I_2 to evolve O_2 . It is necessary to perform mass spectrometry to confirm this. Some quantification (IR, Raman, titration etc.) of LiOH and LiOOH removal are also required in the experiment described in Fig. 3(b), rather than speculating the reaction between hydroxide/peroxide phases and iodine species entirely based on capacities of electrochemistry.

Reply: Thanks for the suggestions. While we are able to confirm the formation of LiOOH and LiOH during the discharging process, we failed to conduct *in-situ* FTIR and Raman spectroscopic measurements to monitor the reactions during charging process. The reason is that the characteristic vibrations of LiOOH unfortunately overlap with that of I_2 in FTIR, while the signals of Raman measurement were too weak to extract meaningful result (see the best result in Figure R6). Based on the fact that both LiOOH and LiOH present as the discharge product, we employed *ex-situ* UV-Vis and mass spectrometry to sequentially monitor the reactions of $\text{LiOOH}\cdot\text{H}_2\text{O} / \text{I}_3^-$ (reaction 5') and LiOH / I_2 (reaction 6'), which are the two reactions proposed for the charging process.

Figure R12. (a) UV-Vis spectra of I_3^- in DME / H_2O (10 / 1) before and after adding excessive LiOOH. (b) UV-Vis spectra of standard 0.1 mM I_3^- solution and the reacted I_2 solution with LiOH. The inset of (b) shows the photos of 6.6 mM I_3^- (1), 10 mM reacted I_2 (2, DME / H_2O = 10 / 1), 10 mM I_2 (3), and 10 mM reacted I_2 (4, DME / H_2O = 1 / 1) solutions.

We firstly investigated the reaction between I_3^- and LiOOH· H_2O . As the UV-Vis spectra shown in Figure R12a, the characteristic absorption peak of I_3^- at 364 nm (extended to the visible region) vanished after mixing with LiOOH· H_2O . As a result, the solution became nearly colourless. Meanwhile, the mass spectrometric measurement in Figure R13a shows that O_2 evolves instantaneously upon mixing I_3^- with LiOOH· H_2O . These observations qualitatively support reaction 5' as indicated below:

The reaction between LiOH and I_2 was also investigated with UV-Vis and mass spectrometry. After adding excessive LiOH and stirring for 1 hour, the colour of I_2 solution in DME/ H_2O (10:1) became slightly lighter (inset of Figure R12b). As the UV-Vis spectra shown in Figure R12b, after reacting with LiOH, the absorption of the I_2 solution became fairly identical to that of the standard solution of I_3^- . That is, the absorption at 400-500 nm was greatly attenuated with only the characteristic peak of I_3^- presented at around 364 nm, which confirms the existence of I_3^- in the reacted I_2 solution. The absorption peak broadening of I_2 solution after titration is presumably attributed to the scattering effect of small particles of LiOH, which has limited solubility in DME/ H_2O (10:1). In addition, considering both I^- and IO_3^- are colourless, $3I_2 + 6LiOH \rightarrow 5LiI + LiIO_3 + 3H_2O$ is unlikely to be the main reaction in DME: H_2O (10:1) solution. So it is more likely that

This is further corroborated with mass spectrometry.

Figure R13. Oxygen evolution recorded from the reactions of (a) LiOOH with I_3^- and (b) LiOH with I_2 monitored by mass spectrometric measurement. The reactions of LiOH with I_2 were performed in solutions with different concentration of H_2O as labelled in the figure.

The mass spectrometric measurement of the reaction between LiOH and I_2 was conducted in two different solutions. As shown in Figure R13b, oxygen evolution was observed instantaneously after I_2 was injected into the 2 M LiOH suspension in DME/ H_2O (10:1). Therefore, O_2 evolution is deemed part of the reaction between I_2 and LiOH in the DME/ H_2O (10:1) electrolyte system, which is the same electrolyte used in the Li- O_2 cell. In contrast, no oxygen was detected when the solvent was changed to DME/ H_2O (1:1). We noticed IO_3^- was detected to be the main product of the reaction between I_2 and LiOH in water-based electrolyte in the literature³,

This reaction does not involve O_2 evolution, our observation is thus consistent with the above reaction. Apparently, the concentration of water is a key factor dictating the reaction between I_2 and LiOH.

(3) The authors need to show more results (IR, Raman, XRD etc) and discuss further on the role of water on the battery chemistry: at what specific water content does the LiOOH- H_2O form in the discharge battery; is there a range of water content, where LiOOH- H_2O forms or not, and why; showing this is essential so that the result can be reproducible. Please state in detail the water content in terms of wt%, vol% and articulate how this water added electrolyte is prepared."

Reply: We thank the reviewer for the suggestions. The formation of LiOOH was investigated by ATR-FTIR measurement via the redox targeting (or ORR) reaction of EV^+ with O_2 in the Li^+ -containing electrolyte of different water content. As the ATR-FTIR spectra shown in Figure R14, the characteristic peaks of both LiOOH and LiOH present in a wide range of water content in the electrolyte (DME/ H_2O (v/v) = 100/1, 100/5, 100/10, and 100/20). Thus the battery chemistry of the discharging process involving LiOOH and LiOH as the main discharge product remains up to water content of 16.7 vol%. We did not attempt to study the reactions at higher water content considering it is not practical for water-contaminated aprotic electrolyte.

For a different purpose, we did observe at 50 vol.% H₂O content (DME/H₂O, 1:1), the OER reaction between LiOH and I₂ follows a distinctly different route with which no O₂ evolution takes place. The results have been shown in Figure R12 and R13. Therefore, we believe the redox targeting (OER) reaction of LiOH·H₂O and I₂ during charging process is dependent on the water content in the electrolyte.

In this work, unless otherwise stated we used volumetric percentage to denote the water content. Water was mixed uniformly with the electrolyte before injected into the cell. After the Li-O₂ cell was assembled, the water-containing electrolyte was introduced into the cell which was then tested in O₂ atmosphere.

A full picture of the battery chemistry requires extensive studies at different conditions (water content, pH, etc.). While we have discovered a different battery chemistry for Li-O₂ cell, we found it is impossible to disclose all the information in a single study. To do that, a systematic study is being planned in future.

Figure R14. ATR-FTIR spectra of LiOH and LiOOH formed from the redox targeting reaction between EV⁺ with O₂ at different concentration of H₂O in the electrolyte.

Reference:

- (1) Brooker, M. H.; Bates, J. B. *The Journal of Chemical Physics* **1971**, *54*, 4788.
- (2) Koura, N.; Kohara, S.; Takeuchi, K.; Takahashi, S.; Curtiss, L. A.; Grimsditch, M.; Saboungi, M.-L. *J Mol Struct* **1996**, *382*, 163.
- (3) Burke, C. M.; Black, R.; Kochetkov, I. R.; Giordani, V.; Addison, D.; Nazar, L. F.; McCloskey, B. D. *ACS Energy Letters* **2016**, 747.
- (4) Bryantsev, V. S.; Giordani, V.; Walker, W.; Blanco, M.; Zecevic, S.; Sasaki, K.; Uddin, J.; Addison, D.; Chase, G. V. *The Journal of Physical Chemistry A* **2011**, *115*, 12399.
- (5) Freunberger, S. A.; Chen, Y.; Peng, Z.; Griffin, J. M.; Hardwick, L. J.; Bardé, F.; Novák, P.; Bruce, P. G. *Journal of the American Chemical Society* **2011**, *133*, 8040.

(6) McCloskey, B.; Bethune, D.; Shelby, R.; Girishkumar, G.; Luntz, A. *The Journal of Physical Chemistry Letters* **2011**, *2*, 1161.

REVIEWERS' COMMENTS:

Reviewer #1 (Remarks to the Author):

The revised manuscript is improved a lot. All of the referee's questions are well answered. It is suitable for publication.

Reviewer #2 (Remarks to the Author):

Comments on the revised paper titled " Proton Enhanced Dynamic Battery Chemistry for Aprotic Li-O₂ Batteries "

This revised manuscript " Proton Enhanced Dynamic Battery Chemistry for Aprotic Li-O₂ Batteries " revised by Wang and coworkers, clearly illustrates a distinct battery chemistry mechanism in water-contaminated aprotic Li-O₂ batteries and LiOOH·H₂O is identified as an unprecedented discharge product, providing a new insights into the understanding of Li-O₂ cell. Moreover, the added or revised experimental results and conclusions are complete and compelling in this revision form, the authors meticulously and accurately reply to the questions which suggested by the reviewer. So the reviewer strongly recommends that this revised manuscript has a qualification for publishing in Nat. Comm. without further revision.

Reviewer #3 (Remarks to the Author):

In the revised manuscript, much more evidence has been shown to support the interesting observation reported. I therefore recommend for publication. Nice work.

Response: We thank the three reviewers for their constructive comments and encouragement.